# Biomarker-Based Responder Selection and Early Prediction of Treatment Response in Hepatocellular Carcinoma: Dynamic Changes in Alpha-Fetoprotein and Des-Gamma-Carboxy Prothrombin During Atezolizumab Plus Bevacizumab Therapy

**DOI:** 10.3390/cancers17243891

**Published:** 2025-12-05

**Authors:** Teiji Kuzuya, Hisanori Muto, Yoshihiko Tachi, Mariko Kobayashi, Hijiri Sugiyama, Mizuki Ariga, Sayaka Morisaki, Gakushi Komura, Takuji Nakano, Hiroyuki Tanaka, Kazunori Nakaoka, Eizaburo Ohno, Kohei Funasaka, Mitsuo Nagasaka, Ryoji Miyahara, Yoshiki Hirooka

**Affiliations:** 1Department of Gastroenterology and Hepatology, Fujita Health University, 1-98 Dengakugakubo, Kutsukake-Cho, Toyoake 470-1192, Aichi, Japan; hisanori.muto@fujita-hu.ac.jp (H.M.); ytachi@fujita-hu.ac.jp (Y.T.); mariko.kobayashi@fujita-hu.ac.jp (M.K.); hijiri.sugiyama@fujita-hu.ac.jp (H.S.); mizuki.ariga@fujita-hu.ac.jp (M.A.); sayaka.kameshima@fujita-hu.ac.jp (S.M.); gakushi.komura@fujita-hu.ac.jp (G.K.); tkjnkn@fujita-hu.ac.jp (T.N.); hiroyuki.tanaka@fujita-hu.ac.jp (H.T.); knakaoka@fujita-hu.ac.jp (K.N.); eizaburo.ono@fujita-hu.ac.jp (E.O.); k-funa@fujita-hu.ac.jp (K.F.); nmitsu@fujita-hu.ac.jp (M.N.); ryoji.miyahara@fujita-hu.ac.jp (R.M.); yoshiki.hirooka@fujita-hu.ac.jp (Y.H.); 2Department of Gastroenterology and Hepatology, Fujita Health University Bantane Hospital, Nagoya 454-8509, Aichi, Japan; 3Department of Gastroenterology, Fujita Health University Okazaki Medical Center, Okazaki 444-0827, Aichi, Japan

**Keywords:** hepatocellular carcinoma, atezolizumab plus bevacizumab, alpha-fetoprotein (AFP), des-gamma-carboxy prothrombin (DCP), tumor markers, immune checkpoint inhibitors (ICIs), early progression, primary non-response, biomarker-based responder selection, progression-free survival

## Abstract

Immune checkpoint inhibitor-based combinations have recently become the standard first-line therapy for unresectable hepatocellular carcinoma. However, some patients derive little or no benefit from the outset, while others who initially achieve disease control later experience early progression. Reliable markers to predict these outcomes remain scarce. This study focused on two widely used tumor markers in clinical practice: alpha-fetoprotein (AFP) and des-gamma-carboxy prothrombin (DCP). We analyzed patients treated with atezolizumab plus bevacizumab and found that early changes in these markers at weeks 3 and 9 accurately predicted both primary progression and early progression after initial disease control. These findings suggest that simple blood tests can support biomarker-based responder selection, providing actionable guidance on whether to continue atezolizumab plus bevacizumab, switch to alternative therapies, or integrate locoregional approaches. Such biomarker-driven adaptive strategies may help personalize treatment and improve outcomes in patients with unresectable hepatocellular carcinoma.

## 1. Introduction

Systemic therapy for unresectable hepatocellular carcinoma (HCC) has advanced markedly in recent years, with immune checkpoint inhibitor (ICI)-based combinations [1,2,3] now established as the standard first-line treatment [4,5,6]. A major clinical challenge, however, is the early identification of patients who show primary progression (1st-PD)—that is, disease progression at the first radiological assessment—so that ineffective therapy can be avoided and subsequent treatment introduced without delay. In this context, the concept of biomarker-based responder selection, using early on-treatment biomarkers to distinguish likely responders from non-responders, has gained increasing attention as a strategy to optimize treatment sequencing and preserve therapeutic opportunities [7].

In addition, a substantial subset of patients who initially achieve disease control (DC) at the first evaluation later display progression at the second evaluation, termed early progression among patients with initial disease control (2nd-PD). Current treatment strategies with ICI-based regimens are increasingly exploring the integration of locoregional therapy with systemic immunotherapy [5,8], even in patients with stable disease (SD), in order to augment antitumor immunity before progression occurs. Identifying patients at risk of 2nd-PD despite initial DC is, therefore, of growing clinical importance, particularly for optimizing treatment sequencing.

Atezolizumab plus bevacizumab (Atz + Bev) has emerged as the predominant first-line ICI-based regimen in contemporary real-world practice following landmark clinical trials [1] and constituted the therapeutic backbone of the present analysis. Concurrently, alpha-fetoprotein (AFP) and des-gamma-carboxy prothrombin (DCP) continue to function as key serum biomarkers in the clinical evaluation and longitudinal monitoring of HCC [9,10], yet reliable biomarkers for predicting treatment response in systemic therapy remain undefined. Recent studies have demonstrated the usefulness of early on-treatment changes in AFP and DCP, particularly around the first radiological evaluation (approximately 6 weeks), for predicting primary progression (1st-PD) [11,12,13,14,15,16]. However, whether these tumor marker dynamics retain predictive value at subsequent evaluations—such as the second imaging assessment—remains unclear.

This study, therefore, aimed to evaluate the predictive value of dynamic changes in AFP and DCP ratios during Atz + Bev therapy in patients with unresectable HCC. Specifically, we investigated the ability of these markers to predict both 1st-PD and 2nd-PD to determine whether serial tumor marker monitoring could serve as a practical tool for biomarker-based responder selection and adaptive treatment optimization in real-world clinical practice.

## 2. Materials and Methods

### 2.1. Patients

Between October 2020 and April 2025, a total of 151 patients diagnosed with unresectable HCC and deemed unsuitable for surgical resection or locoregional therapy received Atz + Bev at our institution. Eligibility criteria were as follows: imaging- or biopsy-confirmed HCC; Barcelona Clinic Liver Cancer (BCLC) stage C or stage B unsuitable for surgery or locoregional therapy; Eastern Cooperative Oncology Group (ECOG) performance status 0–1 (or stable 2); and Child–Pugh score ≤ 7. Prior locoregional therapy, including TACE, and prior systemic therapy were allowed if hepatic function had recovered. Overall, 4 patients with less than six weeks of follow-up were excluded, leaving 147 patients for analysis.

### 2.2. Treatment and Adverse Events

Atezolizumab (1200 mg) and bevacizumab (15 mg/kg) were infused every three weeks. Safety was evaluated using the Common Terminology Criteria for Adverse Events, version 5.0 [17]. If grade ≥ 3 toxicity attributable to treatment developed, administration of the responsible agent(s) was paused and resumed only after improvement to grade 1 or 2. Therapy was maintained unless significant adverse events occurred, disease progression was confirmed, or patients elected to discontinue treatment.

### 2.3. Radiological Assessment

Tumor response was assessed using the Response Evaluation Criteria in Solid Tumors, version 1.1 [18]. Contrast-enhanced computed tomography was performed at baseline, at approximately 6 weeks after initiation of therapy (first evaluation), and thereafter at intervals of 4–10 weeks based on clinical necessity. At the first evaluation (at week 6), patients with complete response (CR), partial response (PR), or SD were classified as the first disease control group (1st-DC), while those with progressive disease (PD) were classified as the 1st-PD group. The second evaluation was defined as the next imaging after the first evaluation and was analyzed using a landmark design restricted to the 1st-DC group.

Patients who were censored before the week 6 evaluation were included in the overall cohort but were not eligible for the week 6 landmark PFS analysis, as they had not reached the predefined landmark time point.

For the second landmark analysis (week 9), only patients who achieved disease control (CR/PR/SD) at the week 6 evaluation and had both AFP and DCP measurements available at approximately week 9 were included. Among the 1st-DC patients, four were excluded from the second evaluation because of insufficient biomarker data or loss to follow-up (including transfer to another institution). Therefore, the week 9 analysis was performed using the remaining patients with complete paired marker data.

### 2.4. Handling of Missing Data and Non-Evaluable (NE) Cases

Missing tumor marker values were not imputed. Patients with very low baseline AFP or DCP levels were excluded from ratio-based analyses because percentage changes cannot be reliably interpreted near the lower detection limit.

Patients who discontinued treatment before the first radiologic assessment due to adverse events or a deterioration of general condition and, therefore, lacked evaluable imaging were classified as non-evaluable (NE). These NE cases were excluded from response-based subgroup analyses (1st-DC and 1st-PD) but were included in the overall survival (OS) and progression-free survival (PFS) analyses until censoring.

### 2.5. Tumor Marker Measurement and Analysis

Serum AFP, DCP, and lens culinaris agglutinin-reactive alpha-fetoprotein (AFP-L3) were measured at baseline and approximately every three weeks during treatment. Patients with baseline AFP ≥ 10 ng/mL, DCP ≥ 40 mAU/mL, or AFP-L3 ≥ 0.5% were included in tumor marker analyses. To evaluate longitudinal changes, each value was normalized to the baseline level (set as 1.0), and the ratio at each time point was calculated. These cutoff values (AFP ≥ 10 ng/mL and DCP ≥ 40 mAU/mL) are conventionally used thresholds indicating clinically meaningful elevations and have been adopted in prior studies evaluating early on-treatment biomarker dynamics in HCC [14]. Patients below these thresholds were excluded from ratio-based analyses to avoid disproportionate percentage changes near the detection limit.

### 2.6. Statistical Analysis

All statistical analyses were performed using EZR version 1.29 (built on R version 3.6.3; Saitama Medical Center, Jichi Medical University, Saitama, Japan) [19]. Progression-free survival (PFS), overall survival (OS), and treatment duration were estimated using the Kaplan–Meier method, and intergroup differences were compared with the log-rank test. In the overall cohort, PFS was defined as the time from initiation of Atz + Bev to radiologic progression or death, whichever occurred first. For landmark analyses, PFS was redefined from the corresponding landmark time points. Specifically, for the week 3 landmark analysis, PFS was measured from the date of the week 3 assessment among patients who remained on treatment and had evaluable AFP and DCP at week 3. For the week 9 landmark analysis, PFS was measured from the week 9 landmark time point among patients in the 1st-DC group who had complete AFP and DCP measurements at approximately week 9. These landmark approaches were adopted to avoid immortal time bias. Categorical variables were examined using Fisher’s exact test. A two-sided *p*-value < 0.05 was considered statistically significant.

The predictive performance of AFP and DCP ratios for identifying primary progression (1st-PD at week 6) and secondary progression (2nd-PD at the second evaluation) was evaluated using receiver operating characteristic curve analysis. The area under the curve was calculated, and optimal cutoff values were derived using the Youden index, along with sensitivity and specificity. For the second evaluation analysis, only patients who achieved 1st-DC at week 6 and had both AFP and DCP available at week 9 were included (landmark design). Kaplan–Meier’s analyses of PFS were performed from the landmark time point. Multivariate Cox’s proportional hazards models were constructed at both the week 3 and week 9 landmark time points to evaluate whether early AFP and DCP dynamics were independently associated with PFS. In these models, AFP and DCP ratios were entered as continuous covariates, and the analyses were adjusted for key clinical factors, including BCLC stage, ECOG performance status, the Child–Pugh score, and treatment line. Hazard ratios (HRs) and 95% confidence intervals (CIs) were calculated. The proportional hazards assumption was checked using Schoenfeld’s residuals, and a *p*-value of <0.05 was considered statistically significant. Because AFP, DCP, and AFP-L3 were considered biologically independent markers, no multiplicity correction was applied to *p*-values. Follow-up was defined from the initiation of Atz + Bev therapy until death or last observation.

### 2.7. Ethical Considerations

This study was approved by the Ethics Committee of Fujita Health University (approval no. HM24-418) and conducted in accordance with the Declaration of Helsinki (1975, as revised in 2013) [20]. Written informed consent for Atz + Bev treatment was obtained from all patients; the requirement for additional consent for study participation was waived by the Ethics Committee due to the retrospective nature of the study.

## 3. Results

### 3.1. Baseline Characteristics

The baseline characteristics of the 147 patients with unresectable HCC treated with Atz + Bev are summarized in Table 1. The median age was 74 years (range, 38–90), and 119 patients (81.0%) were male. Underlying disease etiology included hepatitis B virus infection in 18 patients, hepatitis C virus infection in 29, and non-viral etiologies in 100. The Eastern Cooperative Oncology Group performance status was 0 in 112 patients, 1 in 29, and 2 in 6, while the Child–Pugh scores were 5 in 97 patients, 6 in 37, and 7 in 13.

According to the Barcelona Clinic Liver Cancer (BCLC) classification, 3 patients were stage A, 65 were stage B, and 79 were stage C. Regarding tumor markers, the median AFP level was 50.1 ng/mL (range, 1.8–2,037,310), and 96 patients (65.3%) had AFP ≥ 10 ng/mL. The median DCP level was 613 mAU/mL (range, 10–403,328), with 121 patients (82.3%) showing DCP ≥ 40 mAU/mL. The median AFP-L3 was 16.4% (range, <0.5–99.6), and 117 patients (79.6%) had AFP-L3 ≥ 0.5%.

Atz + Bev was administered as first-line therapy in 106 patients, as second-line in 38, and as third- or later-line therapy in 3. The median observation period was 14.5 months (range, 0.63–53.6).

### 3.2. Overall Treatment Outcomes

Of the 147 patients, the median OS was 21.0 months (95% confidence interval [CI] 17.0–24.6 months), the median PFS was 8.7 months (95% CI 6.7–10.6 months), and the median treatment duration was 6.6 months (95% CI 4.3–7.8 months). The best overall tumor response by RECIST v1.1 was CR in 2 patients (1.4%), PR in 47 (32.0%), SD in 65 (44.2%), PD in 25 (17.0%), and not evaluable (NE) in 9 (6.1%), yielding an overall response rate (ORR) of 33.3% and a disease control rate (DCR) of 77.6%.

#### 3.2.1. Tumor Response at First Evaluation (6 Weeks)

After 6 weeks, radiological evaluation was performed with 138 patients, showing PR in 32 (21.8%), SD in 81 (55.1%), and PD in 25 (17.0%) (Figure 1). The remaining nine patients (6.1%) were NE, mainly due to treatment discontinuation following AEs or a deterioration of general condition, and were excluded from the analyses. Patients with PR or SD were classified as the 1st-DC group (*n* = 113), and those with PD as the 1st-PD group (*n* = 25).

A total of 147 patients with unresectable HCC were enrolled. At the first radiologic evaluation (week 6), the responses were partial response (PR) in 32 patients, stable disease (SD) in 81, progressive disease (PD) in 25, and not evaluable (NE; mainly due to treatment discontinuation or deterioration) in 9. Patients with PR or SD were classified as the 1st-DC group (*n* = 113) and those with PD as the 1st-PD group (*n* = 25). Among the 1st-DC group, 109 proceeded to the second evaluation (median 14.8 weeks): 92 achieved disease control (2nd-DC) and 17 progressed (2nd-PD). Four patients were excluded due to short follow-up or missing data.

#### 3.2.2. Tumor Response at Second Evaluation (Median 14.8 Weeks)

Among the 113 patients in the 1st-DC group, 4 were excluded from the second evaluation for the reasons shown in Figure 1, leaving 109 patients for assessment. The second evaluation was conducted at a median of 14.8 weeks (interquartile range, 14.7–15.7), and all patients were evaluable with no NE cases observed. Of the 30 patients who achieved PR at the first evaluation, 28 maintained PR and 2 experienced PD at the second evaluation. Among the 79 patients who achieved SD at the first evaluation, 14 converted to PR, 50 remained SD, and 15 developed PD. Patients with CR, PR, or SD at the second evaluation were classified as the 2nd-DC group, while those with PD were classified as the 2nd-PD group. As a result, 92 patients were categorized as 2nd-DC and 17 as 2nd-PD.

#### 3.2.3. Tumor Marker Dynamics (1st-DC vs. 1st-PD)

As shown in Table 2 and Figure 2, serum AFP and DCP ratios began to diverge early between the 1st-DC and 1st-PD groups, with significant differences already evident at week 3 (AFP, *p* = 0.0006; DCP, *p* = 0.0046) and further accentuated at week 6. These results indicate that dynamic changes in the ratios at week 3 serve as a practical biomarker-guided tool for predicting primary non-response, while the differences at week 6 reflect consistency with radiological outcomes. In contrast, the AFP-L3 ratio showed no significant differences between the two groups.

#### 3.2.4. Tumor Marker Dynamics (2nd-DC vs. 2nd-PD)

Among patients achieving 1st-DC, AFP and DCP ratios clearly differentiated those who subsequently maintained disease control (2nd-DC) from those who experienced progression (2nd-PD). Significant differences were already evident at week 9 (AFP, *p* = 0.0035; DCP, *p* = 0.0010) and continued to widen at weeks 12 and 15 (Table 3 and Figure 3). From a clinical perspective, week 9 provides the most actionable time point to anticipate early progression despite initial disease control. The AFP-L3 ratio again showed no significant differences between the two groups.

### 3.3. Predictive Performance of Tumor Marker Ratios

We focused on week 3 as the earliest clinically relevant time point for predicting 1st-PD and on week 9 as a landmark time point for predicting 2nd-PD among patients who achieved 1st-DC. These time points were selected a priori to provide actionable clinical information between baseline/first evaluation and the second evaluation.

At the week 3 landmark, ROC analysis identified an AFP ratio ≥1.4 or a DCP ratio ≥ 1.0 as practical thresholds for predicting 1st-PD, with a high sensitivity (0.93) and a negative predictive value (0.97) (Appendix A). In the landmark PFS analysis starting at week 3, patients meeting either cutoff showed significantly shorter PFS compared with those below both thresholds (median PFS from the week 3 landmark: 3.4 vs. 13.1 months; *p* < 0.001) (Figure 4a). For the week 9 landmark analysis in the 1st-DC cohort, an AFP ratio ≥ 1.1 or a DCP ratio ≥ 1.5 provided the best discrimination for predicting 2nd-PD, again with a high sensitivity (0.93) and a negative predictive value (0.97) (Appendix A). Patients above either cutoff experienced markedly shorter PFS than those below both thresholds (median PFS from the week 9 landmark: 3.8 vs. 14.0 months; *p* < 0.001) (Figure 4b).

Together, these findings highlight week 3 as a critical decision point for identifying primary non-responders and week 9 as the optimal landmark for detecting early secondary progression among patients who initially achieved disease control.

To determine whether early AFP and DCP dynamics provided independent predictive value beyond clinical background factors, multivariate Cox’s proportional hazards models were performed at both landmark time points. At the week 3 landmark, the AFP ratio remained a strong independent predictor of PFS (HR 3.822, 95% CI 2.198–6.644; *p* < 0.0001), whereas the DCP ratio did not retain statistical significance after adjustment (Table 4). In contrast, at the week 9 landmark within the 1st-DC cohort, both the AFP ratio (HR 1.725, 95% CI 1.251–2.378; *p* = 0.0009) and the DCP ratio (HR 1.140, 95% CI 1.022–1.273; *p* = 0.0188) remained independently associated with PFS (Table 5). These results indicate that AFP dynamics predominantly drive early prediction at week 3, while both AFP and DCP independently contribute to the identification of early secondary progression at week 9.

## 4. Discussion

This study demonstrated that early dynamic changes in AFP and DCP ratios provide clinically actionable signals for identifying primary progression (1st-PD) at the first evaluation (week 6) and early progression among patients with initial disease control (2nd-PD) at the second evaluation (median: 14.8 weeks). These findings support the application of biomarker-based responder selection to guide adaptive treatment decisions in advanced HCC. To our knowledge, this is the first study to systematically evaluate tumor marker dynamics at both the first and second radiological assessments during ICI-based therapy. Furthermore, we established clinically applicable thresholds at weeks 3 and 9, offering a simple and practical framework for individualized treatment decision-making in patients with unresectable HCC.

Our results confirm and extend previous evidence linking AFP and DCP kinetics with treatment efficacy under ICI-based regimens. At week 3, elevations in AFP and DCP ratios were strongly associated with primary progression. ROC-derived thresholds (AFP ratio 1.41, DCP ratio 0.92) were rounded to AFP ratio ≥ 1.4 or DCP ratio ≥ 1.0 to enhance their clinical usability and avoid misclassification near baseline values. With a high sensitivity (0.93) and a negative predictive value (0.97), this simple rule reliably identifies patients who are unlikely to benefit from continued Atz + Bev therapy. Primary progression under ICI-based therapy is often associated with an “immune-cold” tumor phenotype characterized by poor immune infiltration and limited responsiveness to checkpoint blockades [21,22,23]. For such patients, continued ICI-based therapy is unlikely to confer benefit, and early switching to molecular-targeted therapy—preferably lenvatinib, which uniquely inhibits both VEGFR and FGFR pathways—is recommended in current Japanese treatment algorithms [7,24,25]. Thus, week 3 AFP and DCP dynamics can serve as a cornerstone of biomarker-based responder selection, enabling timely transition to alternative mechanisms of action before irreversible disease progression occurs.

Among patients who achieved disease control at the first evaluation, increases in AFP and DCP ratios from week 9 onward were strongly associated with subsequent progression (2nd-PD). The week 9 thresholds of AFP ratio ≥ 1.1 or DCP ratio ≥ 1.5 serve as clinically actionable indicators to anticipate early progression even before radiographic confirmation; this provides a crucial therapeutic window during which treatment intensification or the integration of locoregional strategies can be implemented while liver function remains preserved. In this context, 2nd-PD may reflect acquired resistance rather than a primary immune-cold phenotype [26,27]. For such patients, early transition to molecular-targeted therapy—preferably lenvatinib—is a rational option [7,25,28]. Alternatively, on-demand locoregional therapy such as transarterial chemoembolization (TACE) targeting less-responsive intrahepatic lesions may enhance tumor antigen release and reinvigorate systemic antitumor immunity, allowing the continuation of the ICI-based regimen (e.g., Atz + Bev) [8,29,30,31]. Such a multimodal approach aims to convert acquired resistance into renewed immune responsiveness and prevent overt progression.

In the present study, multivariate analyses further reinforced the clinical relevance of these biomarker dynamics. At the week 3 landmark, the AFP ratio remained a strong and independent predictor of subsequent PFS after adjustment for BCLC stage, ECOG performance status, the Child–Pugh score, and treatment line, whereas the DCP ratio did not retain significance. This finding suggests that AFP dynamics predominantly capture the biological features underlying primary resistance to immune checkpoint inhibition. In contrast, at the week 9 landmark among patients who initially achieved disease control, both AFP and DCP ratios remained independently associated with PFS, indicating that the combination of these markers reflects emerging acquired resistance during ongoing therapy. These results support the complementary roles of AFP and DCP at different phases of treatment and highlight their utility as non-invasive, dynamic biomarkers for adaptive treatment sequencing.

These findings carry two major clinical implications. Firstly, identifying primary progression as early as week 3 can prevent the futile continuation of Atz + Bev and facilitate earlier sequencing to alternative systemic regimens. Secondly, detecting the risk of early progression at week 9 among 1st-DC patients may enable proactive multimodal strategies—such as on-demand TACE or radiofrequency ablation in combination with ongoing immunotherapy—to enhance antitumor immunity before overt progression occurs [30]. Together, these approaches represent a dual-track treatment framework—early switching for immune-cold primary resistance and immune-enhancing multimodal intervention for acquired resistance—that exemplifies the emerging paradigm of biomarker-based adaptive management in HCC.

Recently, combination strategies integrating systemic therapy and TACE have gained increasing attention, particularly for intermediate-stage HCC [5,8,29]. The integration of ICI-based or anti-VEGF/TKI therapy with TACE may compensate for the limitations of each modality and enhance antitumor immunity. The *REPLACEMENT* study [32] demonstrated that most patients with SD eventually experienced disease progression during atezolizumab plus bevacizumab therapy; therefore, adding TACE before radiological progression to enhance immune activation appears to be a reasonable strategy. Our findings, showing that early AFP and DCP elevations predict subsequent progression among patients with initial disease control, further support this evolving concept by identifying those who may benefit from timely multimodal intervention before radiological progression.

This concept is currently being prospectively tested in the *IMPACT* trial (jRCTs051230037) [33], a randomized phase 3 study in Japan that is evaluating the addition of intrahepatic control TACE to Atz + Bev in patients maintaining stable disease after 6–12 weeks of induction therapy. The *IMPACT* trial aims to determine whether this immune-boosting strategy can prolong overall survival by selectively targeting residual intrahepatic lesions while preserving hepatic function. Notably, early randomization at week 6 is permitted for patients showing tumor enlargement trends or rising AFP levels, reflecting the importance of biomarker dynamics in identifying high-risk SD patients. Our results provide real-world evidence supporting this rationale by demonstrating that increases in AFP or DCP at week 9 predict early secondary progression, indicating a population likely to benefit from such on-demand, immune-enhancing local interventions.

In Japan, recent systemic therapy strategies for unresectable HCC have increasingly emphasized early biomarker-based decision-making during ICI therapy [7,25]. Early changes in AFP and PIVKA-II have been shown to precede imaging-based assessments by 6–12 weeks and to distinguish responders from non-responders within 2–3 weeks after treatment initiation. Our findings expand this concept by demonstrating that dynamic AFP and DCP ratio changes at weeks 3 and 9 during Atz + Bev therapy cannot only identify primary progression but also predict early secondary progression among patients with initial disease control, reinforcing the clinical utility of biomarker-based responder selection in the immunotherapy era. These results complement and extend those of Tanabe et al., who examined early AFP and DCP kinetics primarily in relation to primary progression during Atezo/Bev therapy. While their study highlighted the predictive value of biomarker changes at the initial assessment, it did not evaluate the subsequent clinical course of patients who initially achieved disease control. In contrast, our analysis uniquely focuses on secondary progression (2nd-PD) by assessing biomarker dynamics at the second radiological evaluation using a predefined week 9 landmark. This approach provides new insight into early signals of acquired resistance among patients with initial disease control—an area of major clinical relevance that has not been systematically investigated previously. By integrating both week 3 and week 9 biomarker kinetics, our study proposes a dual-stage framework for identifying primary and secondary non-responders, thereby supporting more precise and timely treatment adaptation during Atezo/Bev therapy.

Although AFP and DCP may provide complementary information, the limited event numbers at the week 3 and week 9 landmark time points preclude the reliable construction of multivariable ROC models or nomograms without a substantial risk of overfitting. Therefore, we adopted simple dual-marker rules to maximize clinical applicability, while future multicenter studies will be needed to establish more complex integrated prediction models.

Although AFP and DCP are widely used tumor markers in HCC, the biological mechanisms linking their early elevation to resistance against immune checkpoint blockades remain incompletely understood. Primary resistance to ICI-based therapy is often associated with an “immune-cold” tumor microenvironment characterized by limited T-cell infiltration, impaired antigen presentation, and high intratumoral heterogeneity. Our findings are consistent with this concept; however, the present study did not include tumor tissue analyses, immune profiling, or circulating tumor DNA. Future studies integrating PD-L1 expression, T-cell infiltration, immune-related gene signatures, or ctDNA dynamics will be necessary to determine whether AFP/DCP kinetics reflect underlying immune phenotypes or molecular features associated with primary or acquired ICI resistance.

This study has several limitations. Firstly, it was a retrospective, single-center analysis with potential selection bias and a relatively small sample size. Secondly, the timing of the second evaluation varied among patients (median, 14.8 weeks), which may have influenced the assessment outcomes. Thirdly, the cutoff values of AFP and DCP ratios at weeks 3 and 9 were determined empirically in this cohort and, therefore, require external validation before clinical application. In addition, because the HBV- and HCV-related subgroups were relatively small, we were unable to reliably assess whether AFP or DCP dynamics differ by underlying disease etiology; larger multicenter cohorts will be needed to evaluate potential etiology-dependent differences in biomarker behavior. We did not perform formal time-dependent ROC analyses because our study was designed around two pre-specified, clinically meaningful landmark time points (weeks 3 and 9), and the number of events after each landmark was limited, making such analyses statistically unstable. Although OS is an important clinical endpoint, this study was specifically designed to evaluate early dynamic biomarker changes at predefined decision points in relation to primary and secondary progression. Because OS is strongly influenced by subsequent systemic or locoregional treatments after progression, it would not directly reflect the predictive value of AFP/DCP kinetics. Moreover, the limited number of OS events after each landmark rendered landmark OS analysis statistically unstable. Therefore, PFS from each landmark was selected as the most appropriate and clinically interpretable endpoint for assessing the utility of early AFP/DCP dynamics. Future studies with larger populations will be required to clarify the impact of these biomarker changes in OS. Finally, the increasing prevalence of tumor marker-negative HCC represents a practical limitation for biomarker-guided approaches. Thus, validation in larger, independent, multicenter cohorts is essential.

## 5. Conclusions

In patients with unresectable HCC treated with Atz + Bev, an AFP ratio ≥ 1.4 or DCP ratio ≥ 1.0 at week 3 reliably identifies primary progression, while an AFP ratio ≥ 1.1 or DCP ratio ≥ 1.5 at week 9 predicts early progression among those who initially achieved disease control. These simple, clinically applicable thresholds enable early treatment switching and the timely integration of locoregional strategies. Such biomarker-based responder selection and adaptive treatment approaches may ultimately improve survival outcomes by ensuring optimal sequencing and preserving future therapeutic opportunities in advanced HCC. Collectively, these findings highlight week 3 as the key time point for identifying primary progression and week 9 as the optimal landmark for predicting early progression, supporting the clinical utility of dynamic AFP and DCP monitoring as a non-invasive and practical biomarker-guided strategy for treatment optimization in unresectable HCC.

## Figures and Tables

**Figure 1 cancers-17-03891-f001:**
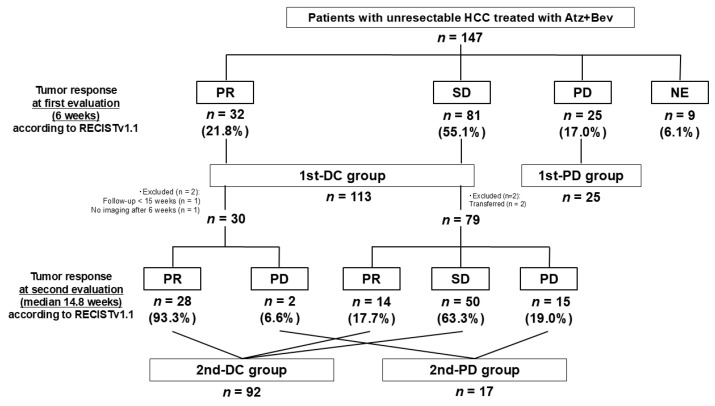
Patient flow diagram and antitumor response during Atz + Bev therapy.

**Figure 2 cancers-17-03891-f002:**
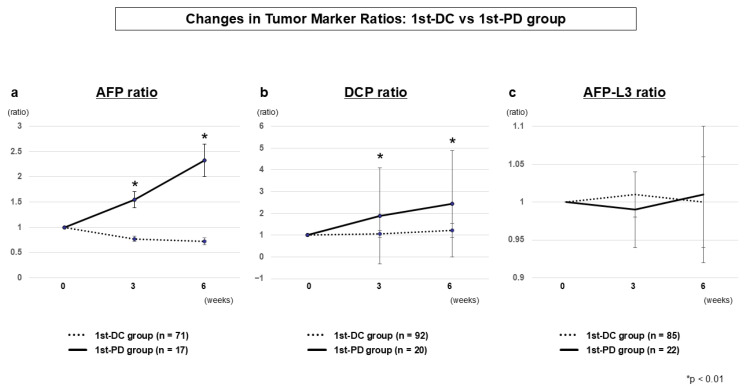
Tumor marker dynamics at the first evaluation (1st-DC vs. 1st-PD) AFP ratio (**a**), DCP ratio (**b**), and AFP-L3 ratio (**c**). AFP and DCP ratios diverged significantly between groups as early as week 3, with clearer separation at week 6. AFP-L3 ratio showed no significant differences.

**Figure 3 cancers-17-03891-f003:**
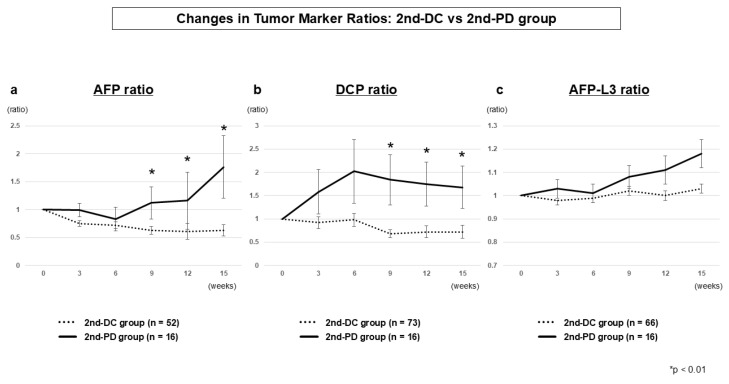
Tumor marker dynamics at second evaluation (2nd-DC vs. 2nd-PD) AFP ratio (**a**), DCP ratio (**b**), and AFP-L3 ratio (**c**). AFP and DCP ratios diverged significantly from week 9 onward, whereas AFP-L3 ratio showed no significant differences.

**Figure 4 cancers-17-03891-f004:**
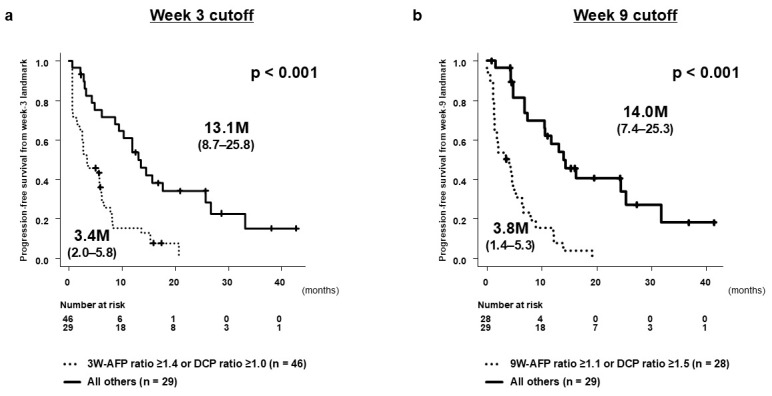
Progression-free survival (PFS) according to early changes in AFP and DCP. (**a**) PFS from the week 3 landmark among patients who remained on treatment and had evaluable baseline and week 3 AFP/DCP measurements. PFS was measured from the week 3 assessment. Patients were stratified according to AFP ratio ≥ 1.4 and/or DCP ratio ≥ 1.0 at week 3. (**b**) PFS from the week 9 landmark among patients in the 1st-DC group who had complete AFP and DCP measurements at approximately week 9. PFS was measured from the week 9 landmark time point. Dashed horizontal lines indicate baseline, and asterisks highlight clinically relevant landmark time points. Statistical details for group comparisons are summarized in the tables.

**Table 1 cancers-17-03891-t001:** Baseline characteristics of patients with unresectable hepatocellular carcinoma treated with atezolizumab plus bevacizumab (Atz + Bev).

Characteristic	
Age (years), median (range)	74 (38–90)
Sex (men/women), *n*	119/28
Etiology (HBV/HCV/non-viral), *n*	18/29/100
ECOG PS (0/1/2), *n*	112/29/6
Child–Pugh score (5/6/7), *n*	97/37/13
BCLC stage (A/B/C), *n*	3/65/79
AFP level (ng/mL), median (range)	50.1 (1.8–2,037,310)
AFP level (<10/≥10 ng/mL), *n*	51/96
DCP level (mAU/mL), median (range)	613 (10–403,328)
DCP level (<40/≥40 mAU/mL), *n*	26/121
AFP-L3 level (%), median (range)	16.4 (<0.5–99.6)
AFP-L3 level (<0.5/≥0.5%), *n*	29/117
Treatment line of Atz + Bev (1st/2nd/3rd/4th), *n*	106/38/2/1
Observation period (months), median (range)	14.5 (0.63–53.6)

Abbreviations: HBV, hepatitis B virus; HCV, hepatitis C virus; non-viral etiology other than HBV or HCV; Atz + Bev, atezolizumab plus bevacizumab; ECOG PS, Eastern Cooperative Oncology Group performance status; BCLC, Barcelona Clinic Liver Cancer; AFP, alpha-fetoprotein; DCP, des-γ-carboxy prothrombin; AFP-L3, lens culinaris agglutinin-reactive alpha-fetoprotein. Note: values are expressed as median (range).

**Table 2 cancers-17-03891-t002:** Tumor marker ratios at weeks 3 and 6 (1st-DC vs. 1st-PD).

Marker	1st-DC Median ± SE (*n*)	1st-PD Median ± SE (*n*)	*p*-Value
AFP ratio 3W	0.77 ± 0.05 (*n* = 71)	1.55 ± 0.16 (*n* = 17)	0.0001
AFP ratio 6W	0.72 ± 0.07 (*n* = 71)	2.33 ± 0.32 (*n* = 16)	<0.0001
DCP ratio 3W	1.05 ± 0.16 (*n* = 92)	1.88 ± 2.21 (*n* = 20)	0.0049
DCP ratio 6W	1.22 ± 0.32 (*n* = 91)	2.44 ± 2.45 (*n* = 19)	0.0022
AFP-L3 ratio 3W	1.01 ± 0.03 (*n* = 85)	1.00 ± 0.06 (*n* = 22)	0.4247
AFP-L3 ratio 6W	0.99 ± 0.05 (*n* = 84)	1.01 ± 0.09 (*n* = 21)	0.1743

Abbreviations: AFP, alpha-fetoprotein; DCP, des-γ-carboxy prothrombin; AFP-L3, lens culinaris agglutinin-reactive alpha-fetoprotein; W, weeks; Atz/Bev, atezolizumab plus bevacizumab; 1st-DC, disease control (complete response, partial response, or stable disease) at first radiological evaluation (6 weeks after Atz/Bev initiation); 1st-PD, progressive disease at first evaluation. Notes: Values are expressed as median ± standard error (SE). *p*-values were obtained using the Mann–Whitney U test. AFP analysis included patients with baseline AFP ≥ 10 ng/mL (*n* = 88); DCP analysis included patients with baseline DCP ≥ 40 mAU/mL (*n* = 113); and AFP-L3 analysis included patients with baseline AFP-L3 ≥ 0.5% (*n* = 117). Large SE values reflect skewed distributions with outliers.

**Table 3 cancers-17-03891-t003:** Tumor marker ratios at weeks 3, 6, 9, 12, and 15 (2nd-DC vs. 2nd-PD).

Marker	2nd-DC Median ± SE (*n*)	2nd-PD Median ± SE (*n*)	*p*-Value
AFP ratio 3W	0.75 ± 0.05 (*n* = 52)	0.99 ± 0.12 (*n* = 16)	0.0192
AFP ratio 6W	0.72 ± 0.06 (*n* = 52)	0.83 ± 0.21 (*n* = 16)	0.0429
AFP ratio 9W	0.63 ± 0.07 (*n* = 52)	1.12 ± 0.29 (*n* = 16)	0.0008
AFP ratio 12W	0.61 ± 0.14 (*n* = 52)	1.16 ± 0.51 (*n* = 16)	0.0003
AFP ratio 15W	0.63 ± 0.10 (*n* = 48)	1.76 ± 0.56 (*n* = 15)	0.0002
DCP ratio 3W	0.92 ± 0.13 (*n* = 73)	1.58 ± 0.48 (*n* = 16)	0.0157
DCP ratio 6W	0.98 ± 0.14 (*n* = 73)	2.02 ± 0.69 (*n* = 16)	0.0178
DCP ratio 9W	0.68 ± 0.09 (*n* = 73)	1.84 ± 0.54 (*n* = 16)	0.0020
DCP ratio 12W	0.72 ± 0.13 (*n* = 71)	1.75 ± 0.47 (*n* = 16)	0.0019
DCP ratio 15W	0.72 ± 0.14 (*n* = 71)	1.68 ± 0.46 (*n* = 15)	0.0020
AFP-L3 ratio 3W	0.98 ± 0.02 (*n* = 66)	1.03 ± 0.04 (*n* = 16)	0.2123
AFP-L3 ratio 6W	0.99 ± 0.02 (*n* = 66)	1.01 ± 0.04 (*n* = 16)	0.2285
AFP-L3 ratio 9W	1.02 ± 0.02 (*n* = 66)	1.08 ± 0.05 (*n* = 16)	0.1448
AFP-L3 ratio 12W	1.00 ± 0.02 (*n* = 66)	1.11 ± 0.06 (*n* = 16)	0.0698
AFP-L3 ratio 15W	1.03 ± 0.02 (*n* = 63)	1.18 ± 0.06 (*n* = 15)	0.0790

Abbreviations: AFP, alpha-fetoprotein; DCP, des-γ-carboxy prothrombin; AFP-L3, lens culinaris agglutinin-reactive alpha-fetoprotein; W, weeks; DC, disease control; PD, progressive disease. Notes: Values are expressed as median ± standard error (SE). *p*-values were obtained using the Mann–Whitney U test (two-sided), without multiple comparison adjustment. Sample size decreased slightly over time due to missing data. AFP-L3 showed no significant differences, although trends toward higher values were observed at weeks 12 and 15. Large SE values reflect skewed distributions with outliers.

**Table 4 cancers-17-03891-t004:** Multivariate Cox’s analysis for progression-free survival from the week 3 landmark.

	Univariate Analysis		Multivariate Analysis	
Factor	HR (95% CI)	*p-*Value	HR (95% CI)	*p-*Value
ECOG PS (0)	0.755 (0.487–1.171)	0.2100	—	—
Child–Pugh score (5)	0.729 (0.495–1.074)	0.1097	—	—
BCLC stage (A or B)	0.896 (0.618–1.298)	0.5613	—	—
AFP ratio 3W (continuous)	3.893 (2.428–6.242)	<0.0001	3.822 (2.198–6.644)	<0.0001
DCP ratio 3W (continuous)	1.065 (1.027–1.104)	0.0006	1.019 (0.972–1.068)	0.4294
Treatment line (first line)	0.694 (0.467–1.031)	0.0708	0.890 (0.494–1.604)	0.6976

**Abbreviations:** AFP: alpha-fetoprotein; DCP: des-gamma-carboxy prothrombin; ECOG PS: Eastern Cooperative Oncology Group performance status; BCLC: Barcelona Clinic Liver Cancer; HR: hazard ratio; CI: confidence interval. **Note:** Cox’s proportional hazards models were constructed at the week 3 landmark among patients who remained on treatment and had evaluable AFP and DCP at week 3. AFP and DCP ratios were entered as continuous variables. Variables with *p* < 0.10 in univariate analysis were included in the multivariate model.

**Table 5 cancers-17-03891-t005:** Multivariate Cox’s analysis for progression-free survival from the week 9 landmark.

	Univariate Analysis		Multivariate Analysis	
Factor	HR (95% CI)	*p-*Value	HR (95% CI)	*p-*Value
ECOG PS (0)	0.875 (0.514–1.491)	0.6232		
Child–Pugh score (5)	0.712 (0.455–1.112)	0.1354		
BCLC stage (A or B)	1.208 (0.782–1.866)	0.3934		
AFP ratio 9W (continuous)	1.929 (1.481–2.512)	<0.0001	1.725 (1.251–2.378)	0.0009
DCP ratio 9W (continuous)	1.155 (1.068–1.249)	0.0003	1.140 (1.022–1.273)	0.0188
Treatment line (first line)	0.888 (0.547–1.443)	0.6315		

**Abbreviations: **AFP: alpha-fetoprotein; DCP: des-gamma-carboxy prothrombin; ECOG PS: Eastern Cooperative Oncology Group performance status; BCLC: Barcelona Clinic Liver Cancer; HR: hazard ratio; CI: confidence interval. **Note:** Cox’s proportional hazards models were constructed at the week 9 landmark among patients who remained on treatment and had evaluable AFP and DCP at week 9. AFP and DCP ratios were entered as continuous variables. Variables with *p* < 0.10 in univariate analysis were included in the multivariate model.

## Data Availability

The data presented in this study are available on request from the corresponding author. The data are not publicly available due to privacy and institutional restrictions.

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
