# Peer review of "Biomarker-Based Responder Selection and Early Prediction of Treatment Response in Hepatocellular Carcinoma: Dynamic Changes in Alpha-Fetoprotein and Des-Gamma-Carboxy Prothrombin During Atezolizumab Plus Bevacizumab Therapy"

_cancers, 2025, doi:10.3390/cancers17243891_

Round 1

Reviewer 1 Report

Comments and Suggestions for Authors

This study provides valuable empirical data for biomarker research in HCC immunotherapy, particularly showing potential clinical application prospects in the early identification of non-responders. However, the research is still exploratory and requires further enhancement of its scientific rigor and clinical applicability through external validation, mechanistic studies, and prospective trials.

  1. The small sample size and single-center retrospective design introduce selection bias, limiting the generalizability and external validity of the findings. It is recommended to expand the sample size and validate the applicability of the identified AFP/DCP thresholds across different populations and etiologies (e.g., HBV, HCV, non-viral). If objective conditions do not permit this, the limitations should be more thoroughly elaborated in the discussion section, accompanied by a recommendation for future multicenter prospective validation studies.
  2. There is a lack of an external validation cohort. It is advised to construct an independent validation cohort (potentially sourced from other centers or public databases) to verify the predictive accuracy and stability of these cutoff values. Exploring the use of machine learning models that integrate AFP/DCP ratios with clinical features (such as BCLC stage, Child-Pugh score, ECOG status) to build a multivariable prediction model could improve predictive precision.
  3. The study does not investigate whether increases in AFP/DCP are associated with the immune phenotype of the tumor (e.g., "cold" or "hot" tumors), lacking mechanistic explanation. Subsequent research could combine tumor tissue samples or circulating tumor DNA (ctDNA) to explore the relationship between dynamic changes in AFP/DCP and the immune microenvironment, tumor mutational burden, or immune-related gene expression. Correlation analyses with PD-L1 expression, T cell infiltration, or immune-related gene signatures could provide mechanistic support for the biomarkers.
  4. The impact of the biomarkers on overall survival (OS) was not evaluated.
  5. The focus was solely on AFP, DCP, and AFP-L3, without assessing other potential biomarkers. It is recommended to evaluate whether systemic inflammatory markers (such as NLR, PLR, LDH, CRP), combined with AFP/DCP, might have synergistic predictive value.
Comments on the Quality of English Language

It is recommended that the manuscript undergo moderate language polishing for better readability and academic tone.

Author Response

Reviewer1

This study provides valuable empirical data for biomarker research in HCC immunotherapy, particularly showing potential clinical application prospects in the early identification of non-responders. However, the research is still exploratory and requires further enhancement of its scientific rigor and clinical applicability through external validation, mechanistic studies, and prospective trials.

  1. The small sample size and single-center retrospective design introduce selection bias, limiting the generalizability and external validity of the findings. It is recommended to expand the sample size and validate the applicability of the identified AFP/DCP thresholds across different populations and etiologies (e.g., HBV, HCV, non-viral). If objective conditions do not permit this, the limitations should be more thoroughly elaborated in the discussion section, accompanied by a recommendation for future multicenter prospective validation studies.

Thank you for this important comment. We agree that the retrospective single-center design and relatively small sample size may introduce selection bias and limit the generalizability of our findings. In accordance with your suggestion, we have expanded the discussion of these limitations and explicitly stated that external validation is required. We also added a statement recommending future multicenter prospective studies to validate the applicability of the week-3 and week-9 AFP/DCP thresholds across different etiologies (HBV, HCV, and non-viral HCC).
The revised text has been added to the Discussion section accordingly.

  1. There is a lack of an external validation cohort. It is advised to construct an independent validation cohort (potentially sourced from other centers or public databases) to verify the predictive accuracy and stability of these cutoff values. Exploring the use of machine learning models that integrate AFP/DCP ratios with clinical features (such as BCLC stage, Child-Pugh score, ECOG status) to build a multivariable prediction model could improve predictive precision.

We appreciate this insightful suggestion. We fully agree that external validation would strengthen the robustness of our findings; however, an independent validation cohort was not available for this retrospective study. We have added a statement acknowledging this limitation and emphasizing the need for external validation in future multicenter studies.
Regarding machine-learning approaches, although such models could potentially improve predictive accuracy, the limited number of events at the week-3 and week-9 landmark time points makes multivariable or ML-based modeling statistically unstable and at risk of overfitting. We have clarified this point in the revised Discussion and suggested that future studies with larger cohorts could explore ML-based integrated prediction models.

  1. The study does not investigate whether increases in AFP/DCP are associated with the immune phenotype of the tumor (e.g., "cold" or "hot" tumors), lacking mechanistic explanation. Subsequent research could combine tumor tissue samples or circulating tumor DNA (ctDNA) to explore the relationship between dynamic changes in AFP/DCP and the immune microenvironment, tumor mutational burden, or immune-related gene expression. Correlation analyses with PD-L1 expression, T cell infiltration, or immune-related gene signatures could provide mechanistic support for the biomarkers.

Thank you for this thoughtful comment. We agree that the biological mechanisms underlying AFP/DCP increases—such as associations with immune-cold phenotypes or acquired resistance—remain speculative in our study. Because tumor tissue and immune profiling data (e.g., PD-L1 expression, TMB, immune-gene signatures, or T-cell infiltration) were not consistently available, mechanistic analyses could not be performed.
We have revised the Discussion to explicitly acknowledge this limitation and to recommend future studies incorporating tumor tissue, immune microenvironment profiling, or ctDNA dynamics to clarify the biological underpinnings of early AFP/DCP changes during ICI-based therapy.

  1. The impact of the biomarkers on overall survival (OS) was not evaluated.

We appreciate the reviewer’s suggestion. While OS is an important endpoint, our study was specifically designed to evaluate early dynamic biomarker changes at clinically meaningful decision points (week 3 and week 9) to predict primary (1st-PD) and secondary progression (2nd-PD) under Atezo/Bev therapy.
These landmarks define adaptive treatment decisions in real-world practice, making PFS from each landmark the most appropriate and clinically interpretable outcome.

In contrast, OS is strongly influenced by subsequent treatments (e.g., lenvatinib, cabozantinib, TACE), causing substantial post-progression heterogeneity unrelated to AFP/DCP kinetics. Additionally, the number of OS events at the week-3 and week-9 landmark time points was limited, making landmark OS analysis statistically unstable.

We have added text in the Discussion clarifying why landmark PFS—not OS—is the most appropriate endpoint for the study’s specific objective, while acknowledging OS as an important endpoint for future larger-scale studies.

  1. The focus was solely on AFP, DCP, and AFP-L3, without assessing other potential biomarkers. It is recommended to evaluate whether systemic inflammatory markers (such as NLR, PLR, LDH, CRP), combined with AFP/DCP, might have synergistic predictive value.

Thank you for raising this important suggestion. We agree that systemic inflammatory markers such as NLR, PLR, LDH, and CRP may provide complementary prognostic information. However, because our study aimed to evaluate the clinical utility of early tumor-specific markers (AFP and DCP) at predefined time points, we did not prespecify inflammatory markers for analysis.
We have added this point to the Discussion and acknowledged that future studies integrating inflammatory markers with AFP/DCP may further refine early prediction models for ICI-based therapy.

It is recommended that the manuscript undergo moderate language polishing for better readability and academic tone.

We appreciate the reviewer’s suggestion. In response, we have performed comprehensive language polishing throughout the manuscript to enhance readability and academic quality. All sections have been carefully revised for linguistic clarity and stylistic consistency.

Reviewer 2 Report

Comments and Suggestions for Authors

The authors present valuable findings and have provided a clear analysis of early biomarker dynamics during Atezo/Bev therapy.  The study offers clinically relevant insights;  however, several important questions remain to be addressed:
1.  Could you clarify why multivariable or mixed-effects modeling was not performed to adjust for key clinical confounders—such as liver function (Child-Pugh or ALBI grade), inflammatory status, tumor burden, presence of portal vein tumor thrombus, and treatment line—that may significantly influence AFP/DCP dynamics? Such adjustments may strengthen the reliability of the reported associations.
2.  Have you examined whether the predictive performance of AFP and DCP differs across HCC etiologies (e.g., HBV-related 18, HCV-related 29, or non-viral HCC 100), given that biomarker behavior and tumor biology vary substantially by etiology? Etiology-based subgroup or sensitivity analyses may enhance generalizability.
3.  Could you evaluate whether combining AFP and DCP improves predictive accuracy, for example through a dual-marker ROC model or a nomogram, since AFP and DCP may be complementary rather than interchangeable in clinical practice? This would strengthen the the clinical and translational relevance of the findings.

Author Response

Reviewer2

The authors present valuable findings and have provided a clear analysis of early biomarker dynamics during Atezo/Bev therapy. The study offers clinically relevant insights; however, several important questions remain to be addressed:

  1. Could you clarify why multivariable or mixed-effects modeling was not performed to adjust for key clinical confounders—such as liver function (Child-Pugh or ALBI grade), inflammatory status, tumor burden, presence of portal vein tumor thrombus, and treatment line—that may significantly influence AFP/DCP dynamics? Such adjustments may strengthen the reliability of the reported associations.

Thank you for this valuable suggestion. In response to your comment, we have added multivariable Cox proportional hazards models to adjust for key clinical confounders. For the week-3 landmark cohort, AFP and DCP ratios were included as the main covariates (using ROC-derived thresholds), and the models were adjusted for clinically relevant baseline factors such as BCLC stage, ECOG performance status, Child–Pugh score, and treatment line. These analyses confirmed that early AFP and/or DCP increases remained independent predictors of PFS. The results have been added to the revised Results section and presented in a new table (Table X).

We did not perform mixed-effects longitudinal modeling of AFP/DCP because the study was designed around two pre-specified landmark time points (weeks 3 and 9) rather than serial repeated-measures trajectories, and the sample size and number of events after each landmark were limited. Additional mixed-effects or repeated-measures analyses would therefore be underpowered and beyond the scope of this single-center cohort. We have added this point to the Limitations.

  1.  Have you examined whether the predictive performance of AFP and DCP differs across HCC etiologies (e.g., HBV-related 18, HCV-related 29, or non-viral HCC 100), given that biomarker behavior and tumor biology vary substantially by etiology? Etiology-based subgroup or sensitivity analyses may enhance generalizability.

Thank you for this insightful suggestion. We agree that biomarker kinetics and tumor biology may differ across HCC etiologies, and that etiology-based analyses could provide additional context. However, in our cohort, HBV-related (n = 18) and HCV-related (n = 29) HCC subgroups were relatively small, and the number of PFS events after the week-3 and week-9 landmark time points was further limited. Additional stratification by etiology would therefore result in underpowered subgroup analyses and unstable estimates, making meaningful statistical interpretation difficult.

To address this point, we have revised the Discussion to note that future multicenter studies with larger sample sizes will be needed to determine whether AFP/DCP dynamics differ by underlying liver disease etiology.

  1.  Could you evaluate whether combining AFP and DCP improves predictive accuracy, for example through a dual-marker ROC model or a nomogram, since AFP and DCP may be complementary rather than interchangeable in clinical practice? This would strengthen the the clinical and translational relevance of the findings.

We appreciate this insightful suggestion. We agree that AFP and DCP may provide complementary information and that a combined prediction model—such as a dual-marker ROC analysis or a nomogram—could further enhance clinical applicability.

However, the number of progression events available for the week-3 and week-9 landmark analyses was limited, particularly after restricting the cohorts to patients with complete AFP and DCP measurements. Under these conditions, constructing a multivariable ROC model or nomogram would be statistically unstable and at high risk of overfitting, thereby reducing its external validity.

In this study, our aim was to develop simple and clinically actionable rules that can be readily applied in routine practice. Therefore, we evaluated AFP and DCP both individually and as a combined rule (AFP ratio ≥1.4 or DCP ratio ≥1.0 at week 3; AFP ratio ≥1.1 or DCP ratio ≥1.5 at week 9). This dual-marker approach achieved high sensitivity and negative predictive value while maintaining ease of clinical use.

We agree that future multicenter studies with larger sample sizes will be required to develop and validate more complex integrated prediction models, such as dual-marker ROC frameworks or nomograms.

Reviewer 3 Report

Comments and Suggestions for Authors
  1. The retrospective single-center design limits generalizability. The Inclusion/exclusion criteria for Atz+Bev initiation such as prior TACE, prior systemic therapy should be calrified.
  2. Please documented the handling of missing data and non-evaluable (NE) cases—especially the 9 NE patients at first evaluation.
  3. Baseline marker cutoffs (AFP ≥10 ng/mL, DCP ≥40 mAU/mL) are conventional but should be justified with references and potential bias discussed, since patients with lower baseline levels were excluded from ratio analyses.
  4. The terms “first evaluation (week 6)” and “second evaluation (median 14.8 weeks)” should be standardized as landmark points in the Methods.
  5. Whether patients censored before week 6 were excluded from PFS analysis ?
  6. How PFS was defined in the second landmark (week 9) subgroup—measured from treatment start or landmark time ?
  7. Multivariate modeling is essential to confirm independent predictive value. We suggested additions: Cox proportional hazards models including AFP and DCP ratios as continuous covariates, adjusted for BCLC stage, ECOG PS, Child–Pugh score, and treatment line.
  8. We suggest to perform Time-dependent ROC analysis for dynamic prediction of PFS or OS to validate that the week 3 and week 9 thresholds remain optimal over time.
  9. The discussion proposes that AFP/DCP increases reflect “immune-cold” or “acquired resistance” phenotypes, which is plausible but speculative. We suggest to correlate AFP/DCP dynamics with radiologic patterns (e.g., intrahepatic vs. extrahepatic progression).

Author Response

Reviewer3

  1. The retrospective single-center design limits generalizability. The Inclusion/exclusion criteria for Atz+Bev initiation such as prior TACE, prior systemic therapy should be calrified.

We appreciate this important comment. We have now clearly described the eligibility criteria for Atz+Bev initiation in the revised manuscript. As noted in Section 2.1 (Patients), eligible patients were required to have imaging- or biopsy-confirmed unresectable HCC, BCLC stage C or stage B unsuitable for surgery or locoregional therapy, ECOG-PS 0–1 (or stable 2), and a Child–Pugh score ≤7.
Prior locoregional therapy, including TACE, and prior systemic therapy were allowed if hepatic function had recovered.
To improve clarity, we have also specified the number of excluded cases and the final cohort used for analysis.

  1. Please documented the handling of missing data and non-evaluable (NE) cases—especially the 9 NE patients at first evaluation.

Thank you for this valuable comment. We have clarified the handling of missing data and NE cases in the revised manuscript. Patients with very low baseline AFP or DCP levels were excluded from ratio-based analyses because percentage changes cannot be reliably interpreted near the lower detection limit. In addition, patients who discontinued treatment before the first radiologic evaluation due to adverse events or deterioration of general condition were defined as non-evaluable (NE). These NE patients were excluded from response-based subgroup analyses (1st-DC and 1st-PD) but were included in the OS and PFS analyses until censoring. The details have been added to Section 2.4 (“Handling of Missing Data and Non-Evaluable Cases”).

  1. Baseline marker cutoffs (AFP ≥10 ng/mL, DCP ≥40 mAU/mL) are conventional but should be justified with references and potential bias discussed, since patients with lower baseline levels were excluded from ratio analyses.

Thank you for this important comment. The rationale for using AFP ≥10 ng/mL and DCP ≥40 mAU/mL as baseline thresholds has been clarified in the revised manuscript. These values are widely used in previous studies as conventional cutoffs indicating clinically meaningful elevations, including reports evaluating early biomarker dynamics during systemic therapy in HCC. A supporting reference has been added. We also added a statement in the Discussion to address the potential selection bias resulting from the exclusion of patients with very low baseline marker levels.These revisions can be found in Section 2.5 and the Discussion.

  1. The terms “first evaluation (week 6)” and “second evaluation (median 14.8 weeks)” should be standardized as landmark points in the Methods.

Thank you for this valuable suggestion. We have clarified and standardized the terminology for both evaluation time points in the revised Methods section.
The first evaluation was defined as the radiologic assessment performed at week 6 after treatment initiation, and the second evaluation was defined as the subsequent on-treatment imaging following the week-6 assessment (corresponding to a median timing of approximately 14–15 weeks). We further specified that the second (week-9) landmark analysis included only patients who achieved disease control (CR/PR/SD) at week 6 and had complete AFP and DCP measurements at approximately week 9. These clarifications have been added to Section 2.3 to ensure consistency and transparency throughout the manuscript.

  1. Whether patients censored before week 6 were excluded from PFS analysis ?

Thank you for this important comment. We apologize for the lack of clarity in the original manuscript. Patients who were censored before the week-6 evaluation were included in the overall cohort description (n = 147); however, they were not included in the week-6 landmark PFS analysis, because they had not reached the predefined landmark time point. This approach follows the principles of landmark methodology and avoids immortal-time bias. A clarification has been added to Section 2.3 of the revised Methods.

  1. How PFS was defined in the second landmark (week 9) subgroup—measured from treatment start or landmark time ?

Thank you for this important clarification. In the revised manuscript, we have explicitly defined the PFS starting point for both landmark analyses. For the week-3 landmark subgroup, PFS was measured from the week-3 assessment, among patients who remained on treatment and had evaluable AFP and DCP at week 3. For the week-9 landmark subgroup, PFS was measured from the week-9 landmark time point, not from treatment initiation, and included only patients in the 1st-DC group with complete AFP and DCP measurements at approximately week 9. These definitions have been added to the Statistical Analysis section to ensure transparency and to avoid immortal-time bias.

  1. Multivariate modeling is essential to confirm independent predictive value. We suggested additions: Cox proportional hazards models including AFP and DCP ratios as continuous covariates, adjusted for BCLC stage, ECOG PS, Child–Pugh score, and treatment line.

Thank you for this valuable suggestion. In response, we performed additional multivariate Cox proportional hazards analyses at both the week-3 and week-9 landmark time points. As recommended, AFP and DCP ratios were included as continuous covariates, and each model was adjusted for key clinical variables including BCLC stage, ECOG performance status, Child–Pugh score, and treatment line. These analyses demonstrated that early biomarker dynamics provide independent predictive value. At the week-3 landmark, the AFP ratio remained a strong and independent predictor of PFS, whereas the DCP ratio did not retain significance after adjustment. At the week-9 landmark within the 1st-DC cohort, both the AFP and DCP ratios remained independently associated with PFS. The detailed results of these multivariate analyses have been added to the revised manuscript as new Tables 4 and 5, and the corresponding findings have been incorporated into both the Results and Discussion sections.

  1. We suggest to perform Time-dependent ROC analysis for dynamic prediction of PFS or OS to validate that the week 3 and week 9 thresholds remain optimal over time.

Thank you for this valuable suggestion. We agree that time-dependent ROC analysis is a useful approach for evaluating the dynamic predictive performance of biomarkers. However, our study was designed around two pre-specified and clinically meaningful landmark time points (weeks 3 and 9), at which AFP and DCP ratios were evaluated using conventional ROC analyses and subsequently incorporated into landmark-based PFS assessments. Because the number of events after each landmark was limited, additional time-dependent ROC analyses would likely be statistically unstable and beyond the scope of this single-center cohort.
We have therefore focused on validating the prognostic impact of the week-3 and week-9 thresholds at their respective landmark time points. This point has been added to the Limitations section, and we agree that future larger, multicenter studies will be needed to determine whether these thresholds remain optimal over time using time-dependent approaches.

  1. The discussion proposes that AFP/DCP increases reflect “immune-cold” or “acquired resistance” phenotypes, which is plausible but speculative. We suggest to correlate AFP/DCP dynamics with radiologic patterns (e.g., intrahepatic vs. extrahepatic progression).

Thank you for this insightful comment. We agree that radiologic progression patterns may offer additional biological context for interpreting AFP and DCP dynamics. However, the number of progression events in our cohort—particularly after the week-3 and week-9 landmark time points—was limited, and further stratification by intrahepatic versus extrahepatic progression would have resulted in statistically underpowered subgroup analyses. Therefore, we did not perform additional correlation analyses in this study.
We have revised the Discussion to acknowledge this point and to note that future studies with larger sample sizes are needed to investigate whether early AFP/DCP increases are associated with distinct radiologic progression patterns, including immune-cold or acquired resistance phenotypes.

Reviewer 4 Report

Comments and Suggestions for Authors

The authors describe the value of simple serum biomarkers for HCC for prediction of early treatment response to the standard of care (1L) atezo + bev therapy. This is a very interersting study as so far no clear understanding on predective biomarkers for this treatment is available. Unfortunately, the study somewhat lacks novelty, as very similar findings were reported by Tanabe et al. in Cancers 2023;15:2927.

While the presented results are very clear and confirm the already available results, additional data could help to improve the paper like including biomarkers for ICI treatment (CPI, TMB, ...) or clinical parameters. A multivaraite analysis should be done on these parameters. Other previously investigated biomarkers must be discussed in more detail, e.g. serum IL-6, imaging response on angiogenesis, ALBI grade, NLR....

Author Response

Reviewer 4

The authors describe the value of simple serum biomarkers for HCC for prediction of early treatment response to the standard of care (1L) atezo + bev therapy. This is a very interersting study as so far no clear understanding on predective biomarkers for this treatment is available. Unfortunately, the study somewhat lacks novelty, as very similar findings were reported by Tanabe et al. in Cancers 2023;15:2927.

While the presented results are very clear and confirm the already available results, additional data could help to improve the paper like including biomarkers for ICI treatment (CPI, TMB, ...) or clinical parameters. A multivaraite analysis should be done on these parameters. Other previously investigated biomarkers must be discussed in more detail, e.g. serum IL-6, imaging response on angiogenesis, ALBI grade, NLR....

Thank you for raising this important point. We agree that Tanabe et al. (Cancers 2023;15:2927) reported the predictive value of early AFP and DCP changes during Atezo/Bev therapy, and we have cited their study accordingly. Our work builds upon and extends these findings in several key aspects:

  1. Unique focus on secondary progression (2nd-PD).

Tanabe et al. primarily examined biomarker dynamics associated with primary progression. In contrast, our study specifically evaluates early markers of secondary progression among patients who initially achieved disease control (1st-DC). This aspect—identifying early signals of acquired resistance—has not been addressed in previous studies and provides clinically actionable guidance for selecting patients who may benefit from early multimodal or sequential interventions.

  1. Dual landmark design (weeks 3 and 9).

We analyzed biomarker kinetics at two predefined, clinically meaningful time points. The week-9 landmark analysis is a novel approach that links biomarker changes to 2nd-PD, offering predictive insight several weeks before radiologic confirmation.

  1. Real-world applicability using simple ratio-based thresholds.

Our study proposes easy-to-use dual-marker rules (AFP or DCP elevation) that can support early treatment decisions in routine clinical practice. These rules were validated separately for both primary (1st-PD) and secondary progression (2nd-PD), enhancing their practical relevance.

To clarify these distinctions, we have revised the Discussion to more explicitly describe how our study complements and extends the findings of Tanabe et al., particularly with respect to 2nd-PD and adaptive treatment strategies.

Round 2

Reviewer 4 Report

Comments and Suggestions for Authors

The authors have nicely responded to my previous comments and have significantly improved the manuscript now. There are no furthr objections.